# Effect of *Spirulina* Supplementation on Systolic and Diastolic Blood Pressure: Systematic Review and Meta-Analysis of Randomized Controlled Trials

**DOI:** 10.3390/nu13093054

**Published:** 2021-08-31

**Authors:** Piotr Machowiec, Gabriela Ręka, Marcela Maksymowicz, Halina Piecewicz-Szczęsna, Agata Smoleń

**Affiliations:** Department of Epidemiology and Clinical Research Methodology, Medical University of Lublin, 20-080 Lublin, Poland; gabrysia.reka@gmail.com (G.R.); marcela.maksymowicz@gmail.com (M.M.); halpiec@gmail.com (H.P.-S.); agata.smolen@umlub.pl (A.S.)

**Keywords:** *Spirulina*, blood pressure, diastolic pressure, systolic pressure, hypertension, food supplements, meta-analysis

## Abstract

*Spirulina* is a microalga that presents various important pro-health properties, for instance lowering blood pressure in the research. The study aims to appraise the efficacy of *Spirulina* administration on systolic (SBP) and diastolic blood pressure (DBP). Randomized controlled trials (RCTs) were retrieved by a systematic search of PubMed, Web of Science, and the Cochrane Library databases from inception to June 2021 according to a standardized protocol. The effect size of each study was counted from mean and standard deviation before and after the intervention and shown as Un-standardized mean difference and 95% confidence interval. Sensitivity analyses were performed. Meta-analysis on 5 RCTs with 230 subjects was eligible. The amount of *Spirulina* ranged from 1 to 8 g per day, and intervention durations ranged from 2 to 12 weeks. Data analysis indicated that *Spirulina* supplementation led to a significant lowering of SBP (Mean Difference (MD): −4.59 mmHg, 95% Confidence Interval (CI): −8.20 to −0.99, I square statistic (I^2^) = 65%) and significant lowering of DBP (MD: −7.02 mmHg, CI: −8.86 to −5.18, I^2^ = 11%), particularly in a subgroup of hypertensive patients. *Spirulina* administration might have a supportive effect on the prevention and treatment of hypertension. More exact randomized controlled trials are needed to clarify the effect of *Spirulina* supplementation on blood pressure.

## 1. Introduction

Arterial hypertension is the crucial independent risk factor in cardiovascular diseases for developed countries, characterized by an increasing prevalence in recent years [1,2]. Despite significant advancement in perception, diagnosis, and therapy of hypertension, it is underlined that blood pressure control is insufficient in less than half of the hypertensive patients and it poses a challenge for further patients’ management [2]. The most common method of treatment of hypertension is drug therapy. Nevertheless, one of the difficulties with decreasing blood pressure values is that monotherapy is often inadequate and that is why combination therapy may be considered in some cases [3,4], especially in elderly patients. Alternatively, natural medical products are frequently used as adjunctive therapy to improve long-term outcomes in patients with arterial hypertension and to reduce the number of antihypertensive drugs that are taken. Thus, it may potentially exclude the side effects concerned with combination therapy use [5].

The intake of microalgae has been popularized for centuries and for the sake of their nutritional value and properties, its intake is recommended by organizations, such as WHO (World Health Organization) and FAO (The Food and Agriculture Organization) [6,7]. Microalgae contain a variety of biologically valuable substances including proteins, polyunsaturated fatty acids, amino acids, antioxidants, pigments, minerals, and vitamins [7]. *Spirulina*, a microscopic and filamentous cyanobacterium is considered a sustainable and eco-friendly microalga, playing an increasing role in alternative medicine [8].

*Spirulina* presents various important pro-health properties according to experimental studies and human research. Antioxidant, anti-inflammatory, and immunomodulatory activities of *Spirulina* were investigated in experimental studies on animals [9,10,11,12]. Hypoglycemic and hypolipidemic activities were described in human research [9,13]. According to clinical studies, the antioxidant activity of *Spirulina* might be a potential treatment for chronic obstructive pulmonary disease and might improve motor development among infants [10,14].

The effect of *Spirulina* intake on blood pressure is considered in some human randomized clinical trials. However, the results are not completely decisive and they should be deepened. A few studies indicated that *Spirulina* significantly changes (lowers) only the systolic blood pressure (SBP) [15,16] or only the diastolic blood pressure (DBP) [17]. Two studies reported that there were no statistically significant differences between the experimental and control group regarding systolic and diastolic blood pressure [18,19]. In this meta-analysis, we evaluated the effectiveness of *Spirulina* supplementation on systolic and diastolic blood pressure to elucidate the subject. According to the PICOS (Participants, Intervention, Comparison, Outcomes, Study design) statement, our meta-analysis aims to find an answer to the question does the oral use of *Spirulina* in a daily dose from 1 to 8 g and more for 2, 8, or 12 weeks in adult patients with co-existing diseases reduce systolic and diastolic blood pressure.

## 2. Materials and Methods

### 2.1. Review Design and Search Strategy

The Preferred Reporting Items for Systematic Reviews and Meta-Analyses (PRISMA) guideline was complied with to perform a meta-analysis and report the results [20]. Three databases including PubMed, Web of Science, and the Cochrane Library from inception to June 2021 were systematically searched to find accurate studies. The search terms for the PubMed database were: (*Spirulina*) AND (intervention studies [MESH] OR intervention [tiab] OR controlled trial [tiab] OR random [tiab] OR randomised [tiab] OR randomized [tiab] OR randomly [tiab] OR assignment [tiab] OR clinical trial [All fields] OR trial [All field]) and for searching the Cochrane Library: *Spirulina* AND (clinical trial OR trial OR clinical study). The following phrases were used to search Web of Science: (*Spirulina*) AND (intervention studies OR intervention OR controlled trial OR random OR randomised OR randomized OR randomly OR assignment OR clinical trial OR trial). Manual searching of retrieved studies was performed not to miss randomized controlled trials (RCTs). Two authors (GR & MM) independently explored databases and carefully evaluated the articles. Inconsistent issues were resolved by discussion with the other two authors (PM & HPS).

### 2.2. Study Selection

Typical inclusion criteria used for the selection of all relevant articles were as followed: (i) the parallel-group or cross-over RCTs in which *Spirulina* was administered in the intervention group, (ii) RCTs used a concurrent control group for the *Spirulina* supplementation group, and the difference between intervention and control group was *Spirulina*, (iii) RCTs provided sufficient data on the baseline and final levels of blood pressure, both SBP and DBP in *Spirulina* and control group. The search was restricted to English-language publications (iv). Non-randomized, non-control, experimental studies and studies with a lack of adequate data on SBP and DBP values before and after *Spirulina* supplementation required for meta-analysis were excluded. Publications without clear information on the selection of patients and unambiguous method of carrying out randomization and blinding process were rejected, which could have contributed to overestimating the effect of the intervention.

### 2.3. Extraction and Qualification of the Data

For each selected RCTs, the following data were extracted: first author’ identification, publication year, study design, study location, number of participants in *Spirulina* and control groups, gender and mean age of subjects, participants’ health status, daily dose, intervention duration, and outcomes of interest (SDB and DBP levels).

To evaluate the quality of included RCTs, the Jadad scale was used [21]. The scores ≥ 3 were classified as high-quality trials, whereas scores < 3 as low-quality trials (possible score ranges between 0 and 5) [22]. Two authors (PM & GR) independently classified the included studies according to the Jadad scale. To make a consensus, final scores were consulted with the authors.

### 2.4. Statistical Methods

The effect size of each study was counted from the mean and standard deviation (SD) of the results before and after the intervention, and subsequently shown as Un-standardized mean difference and 95% confidence interval (CI). To extract one mean and one standard deviation in the studies, in which mean ± SD were divided into females and males, the certain transformations were used based on Cochrane’s formulae for combining groups (Table 1). Standard deviations (SDs) of the mean difference between pre-treatment and post-treatment values in the intervention and control groups were calculated using the following formula: SD = square root [(SD_pre-treatment_)^2^ + (SD_post-treatment_)^2^ − (2R × SD_pre-treatment_ × SD_post-treatment_)], imputing a correlation coefficient as described in Cochrane recommendations (Table 1). In studies where the standard error of the mean (SEM) was reported, SD was calculated as follows: SD = SEM × sqrt (n), where n is the number of participants. On account of the fact that selected RCTs were carried out in different settings including the type of *Spirulina* supplement used, *Spirulina* dose, duration of *Spirulina* supplementation, and demographic characteristics of in-dividual trials, random effects model was used to calculate the overall effect from effect sizes [23,24]. The effect of heterogeneity was quantified by I-squared (I^2^) statistic, ranging from 0 to 100% as I^2^ > 60% refers to considerable heterogeneity [25]. To find the potential sources of between-study heterogeneity, we carried out a pre-planned subgroup analysis based on dose of intervention, duration of *Spirulina* intake and baseline blood pressure. In case of significant heterogeneity random effect analysis was performed. Sensitivity analysis was carried out to explore the inference of each study on overall effect. Potential publication bias was not assessed due to an insufficient number of studies (*n* < 10), so the power of the tests is too low to differentiate chance from real asymmetry [26]. Review Manager software version 5.4.1 (Cochrane IMS, Oxford, UK) was employed to perform statistical analyses and draw the forest plots. *p* values ≤ 0.05 was considered as statistically significant.

## 3. Results

### 3.1. Selection and Identification of Studies

From database searching, 774 records were retrieved in our systematic search. Of these, 233 duplicated records were excluded. Among the remaining, 533 were removed. Reasons for exclusion present as follows: 441 items irrelevant to the topic, 52 trial protocols, 6 animal/non-human studies, 20 review articles and other non-RCTs, and 14 due to lack of data on systolic/diastolic blood pressure values before or after *Spirulina* supplementation. The studies containing only the values of SBP or DBP were accepted solely when comprising pre- and post *Spirulina* supplementation blood pressure values. The next 3 records were rejected because they did not report the results. Finally, 5 studies were available for the main analysis. A flow diagram showing the accurate study selection and identification process is presented in Figure 1.

### 3.2. Characteristics of Studies

In total, 5 randomized controlled trials with 230 subjects for SBP and 4 RTCs with 214 subjects for DBP were enrolled in meta-analysis [15,16,17,18,19]. PICOS criteria for inclusion and exclusion of studies are available in Table 2. All studies were published between 2008 and 2021 and were conducted in five different countries, including Mexico, Poland, South Korea, the USA, and Iran (Table 3). All RCTs were designed as parallel-group studies. Four of them used the placebo as the control and one trial used no intervention as the control. The dose of *Spirulina* ranged from 1 to 8 g per day. The intervention duration was 3 months (12 weeks) for three studies, 8 weeks for one study, and 2 weeks for another. All patients were adults and they suffered from co-existing diseases, for instance, hypertension, diabetes mellitus type 2, chronic joint pain, and ulcerative colitis (Table 4).

According to Jadad scores [21], 1 study [15] was categorized as a low-quality study (score < 3) and 4 studies [16,17,18,19] as high-quality studies (score ≥ 3).

Effect sizes for the impact of *Spirulina* supplementation on systolic blood pressure were robust in the sensitivity analysis. It suggests that desertion of each RCT had no significant effect on the results. Also, sensitivity analysis for diastolic blood pressure was conducted, indicating primarily that dropping one study influenced overall effect significantly. However, after the exclusion of one trial, sensitivity analysis showed the same results as in the case of SBP.

Two authors (HPS & AS) separately assessed the methodological quality of RCTs through Cochrane Collaboration’s tool which includes six domains (Figure 2). Each domain was classified to the following categories: low, high, and unclear risk of bias (1,0, or?). Overall quality of each study was considered as good, fair, and weak: >2, =2, and <2, respectively. After evaluating the quality of included studies, the quality score of 4 studies were classified as good quality and 1 as fair quality.

### 3.3. Effect of Spirulina on Systolic Blood Pressure

The effect of the *Spirulina* administration on systolic blood pressure was assessed based on 5 RCTs. Pooled analysis revealed that *Spirulina* intake led to a significant lowering of SBP (MD: −4.59 mmHg, 95% CI: −8.20 to −0.99, I^2^ = 65%). There was a significant between-study heterogeneity (I^2^ = 65%; *p* = 0.02) in which dose of *Spirulina*, duration of *Spirulina* intake, and baseline blood pressure were identified as sources of heterogeneity in the subgroup analysis (Figure 3). As shown in Figure 4, subgroup analysis based on a dose of supplementation indicated no difference between low-dose—≤2 g (MD: −4.17 mmHg, 95% CI: −9.85 to 1.52) or high-dose—>2 g (MD: −5.17 mmHg, 95% CI: −12.60 to 2.26) supplementation. Furthermore, subgroup analysis based on “≥12 weeks” or “<12 weeks” duration of *Spirulina* intake did not show any difference between subgroups (MD: −6.67 mmHg, 95% CI: −11.53 to −1.81 vs. MD: −0.87 mmHg, 95% CI: −4.77 to 3.03) (Figure 5). The only significant difference was noticed in subgroup analysis based on baseline blood pressure. It indicated that *Spirulina* supplementation resulted in greater SBP lowering in the “hypertensive” subgroup (MD: −9.18 mmHg, 95% CI: −14.93 to −3.43) compared with the “normotensive” subgroup (MD: −2.26 mmHg, 95% CI: −4.44 to −0.08) (Figure 6).

### 3.4. Effect of Spirulina on Diastolic Blood Pressure

Pooled analysis of 4 RCTs showed that *Spirulina* intake significantly lowered DBP (MD: −4.29 mmHg, CI: −8.43 to −0.14, I^2^ = 81%) in comparison with the control group. However, the above analysis showed unacceptable heterogeneity (I^2^ = 81%). Therefore, based on the sensitivity analysis, we found and eliminated the study [19] that caused high heterogeneity (Figure 7a). After re-analysis, which accounted for the exclusion of the above study, we noticed highly significant lowering of DBP in the *Spirulina* group (MD: −7.02 mmHg, CI: −8.86 to −5.18, I^2^ = 11%) compared with the control group (Figure 7b).

## 4. Discussion

Arterial hypertension is one of the most common health problems in developed countries and is a risk factor for cardiovascular diseases. Numerous studies have shown the positive effect of certain nutrients and dietary interventions on high blood pressure levels [27]. Diet with natural fruits and vegetables containing antioxidants has a blood pressure-lowering effect [27]. According to recent studies, lycopene-carotenoid from tomatoes or such nutraceuticals as flavonoids contained in cacao, beetroot with nitrates, garlic, and fish oil, being a source of unsaturated fats, have the potential to improve blood pressure [27,28].

In this paper, we present the results of a systematic review and meta-analysis which indicated *Spirulina* supplementation significant reduction of systolic and diastolic blood pressure. Our meta-analysis focuses on SBP and DBP parameters and not all components of metabolic syndrome, which makes the study comprehensible. *Spirulina* is a functional food that might have a beneficial effect on decreasing blood pressure. However, individual RCTs showed some different results regarding blood pressure values between themselves. Martínez-Sámano et al. showed a statistically significant decrease in systolic blood pressure after 12 weeks of *Spirulina* supplementation in a dose of 4.5 g per day (*p* < 0.05) and no statistically significant changes regarding the diastolic blood pressure [15]. Miczke et al. confirmed a hypotensive effect of *Spirulina*. Significant decreases in SBP and DBP were observed in the *Spirulina* group after three months of treatment with 2 g [16]. Administration of 8 g per day of *Spirulina* for 12 weeks showed a significant lowering effect on DBP (*p* < 0.021) and no significant effect on SBP in the study by Lee et al. [17]. Results of the study by Jensen et al. indicated that with a dose of 2.3 g there were no statistically significant differences between *Spirulina* and placebo groups regarding SBP and DBP at baseline or after 2 weeks. However, the consumption was associated with a mild reduction in DBP [18]. In contrast, Moradi et al. claimed that *Spirulina* taken 1 g per day by ulcerative colitis patients did not influence blood pressure values (*p* > 0.05) [19].

Discrepancies require further in-depth analyzes. Some other possible confounding factors in assessing the impact of *Spirulina* on SBP and DBP, for example, physical activity, diet, and smoking should be taken into consideration. These conflicting outcomes of the studies can depend on the supplementation of different doses of *Spirulina* in different periods of time. The fact that no significant hypotensive effect was seen after administration of 1 g of *Spirulina* might suggest the role of the appropriate dosage of the supplement to decrease blood pressure. It is worth mentioning that no significant changes in blood pressure parameters were noted in studies, in which the duration of *Spirulina* supplementation was below 12 weeks: 2 or 8 weeks, respectively [18,19]. Studies of 12 weeks of *Spirulina* administration showed a significant decrease in at least one component of blood pressure [15,16,17]. However, our subgroup analysis based on a dose of supplementation indicated that there was no significant difference between dose ≤2 g or >2 g, and subgroup analysis based on “≥12 weeks” or “<12 weeks” duration of *Spirulina* intake did not show any difference between subgroups. Meta-analysis results are evidence of greater importance than analysis of particular studies. It is worth highlighting that *Spirulina* supplementation resulted in greater SBP lowering in a subgroup of hypertensive patients compared with those with normal blood pressure. It suggests that patients with hypertension might reap greater benefits with *Spirulina* supplementation. Both dosage and the duration of supplementation require further studies.

Other non-randomized controlled studies assessing a relationship between *Spirulina* intake and blood pressure values showed that 4.5 g per day for 6 weeks had the positive effect on SBP and DBP reduction in a sample of overweight patients [29]. On the contrary, Mazopakis et al. found no significant changes in SBP and DBP after the intervention of 1 g of *Spirulina* per day for 12 weeks in a Cretan population [30,31].

The mechanism of lowering blood pressure by *Spirulina* is partially understood. It was supposed that the high content of potassium in *Spirulina* might have a lowering effect on blood pressure [17]. Phycocyanin, a blue pigment with antioxidant activity from *Spirulina*, decreases parameters of blood pressure by strenghtening the expression of endothelial nitric oxide synthase in the aorta after the stimulation of adiponectin [19,32]. Oxidative stress connected to endothelial damage, contributing to a decrease in nitric oxide synthase (NOs), and decreased vasoconstriction has been reported in hypertension [15]. In mice, the decameric peptide of *Spirulina platensis* decreases blood pressure levels through a PI3K (phosphoinositide-3-kinase)/AKT (serine/threonine kinase Akt)/eNOS (endothelial NO synthase) -dependent mechanism [33]. Martínez-Sámano et al. proved the antioxidative properties of *Spirulina* in SBP—they observed an increase in glutathione peroxidase (GPx) activity and oxidized glutathione (GSSG) concentrations (*p* < 0.05) [15]. Additionally, sVCAM-1, sE-selectin, and endothelin-1 levels—considered as markers of endothelial dysfunction were reduced [15]. *Spirulina* improves endothelial function by reducing arterial stiffness index (SI) [34]. Moreover, *Spirulina* contains natural angiotensin I converting enzyme inhibitor (ACEi) peptide. ACEi suppresses the synthesis of angiotensin II that induces the vasoconstriction of blood vessels and the release of aldosterone, resulting in blood pressure increase [34,35,36]. More studies should be performed to confirm these hypotheses and evaluate the exact mechanism of antihypertensive properties of the substance in humans.

Due to many complications of hypertension, intake of *Spirulina* with antioxidant and hypotensive activity might reduce blood pressure, which potentially reduces cardiovascular risk and prevents serious effects such as stroke or heart attack. Moreover, hypertension is frequently associated with diabetes mellitus and metabolic syndrome. Some studies revealed that *Spirulina* intake improved glucose and lipid metabolism, reduced oxidative stress, modulated appetite, so it can be considered as a therapeutic nutraceutical not only by reducing blood pressure [13,32]. According to our analysis, *Spirulina* may potentially reduce blood pressure among hypertensive patients. Supplementation of *Spirulina* products promoted as “superfoods” is more and more popular due to its health benefits, but recent studies revealed contamination of toxic substances—cyanotoxins, heavy metals, or polycyclic aromatic hydrocarbons (PAHs) [37,38]. Further research is needed, which doses and forms are the most effective and safe for patients. It is necessary to assess the safety profile of combination therapy consisting of *Spirulina* and pharmacotherapy.

The limitations of this meta-analysis were the small number of analyzed RCTs and the lack of possibility to assess publication bias. However, the greatest strength of this paper is the inclusion of two novel studies from 2018 and 2021 [15,19]. The studies were first included in a meta-analysis that broadened the analysis compared with the previous meta-analysis assessing the influence of *Spirulina* supplementation on a decrease of blood pressure. In the meta-analysis by Huang H. et al. *Spirulina* supplements significantly lowered DBP (weighted mean differences = −7.17 mmHg; 95% CI: −8.57 to −5.78; *p* = 0.0001; I^2^ = 0%), but not SBP (weighted mean differences = −3.49 mmHg; 95% CI: −7.19 to 0.21; *p* = 0.06; I^2^ = 50%) [39]. Differences in included studies might be a reason for different results for SBP compared with our meta-analysis. Yousefi et al. in a systematic review came to similar conclusions to our meta-analysis that additional studies with greater sample sizes and extended durations are needed to establish the hypotensive effect of *Spirulina* [40].

## 5. Conclusions

In summary, the results of the current meta-analysis indicated that *Spirulina* supplementation might have a supportive effect on the prevention and treatment of hypertension. *Spirulina* intake had beneficial hypotensive properties, particularly in patients with hypertension compared to those with normal blood pressure. *Spirulina* should be further used as a nutraceutical food supplement due to its pro-health properties. More RTCs with different dosages and the duration of *Spirulina* administration are recommended to be conducted to clarify the effect of *Spirulina* on systolic and diastolic blood pressure and to determine its clinical relevance in hypertension management.

## Figures and Tables

**Figure 1 nutrients-13-03054-f001:**
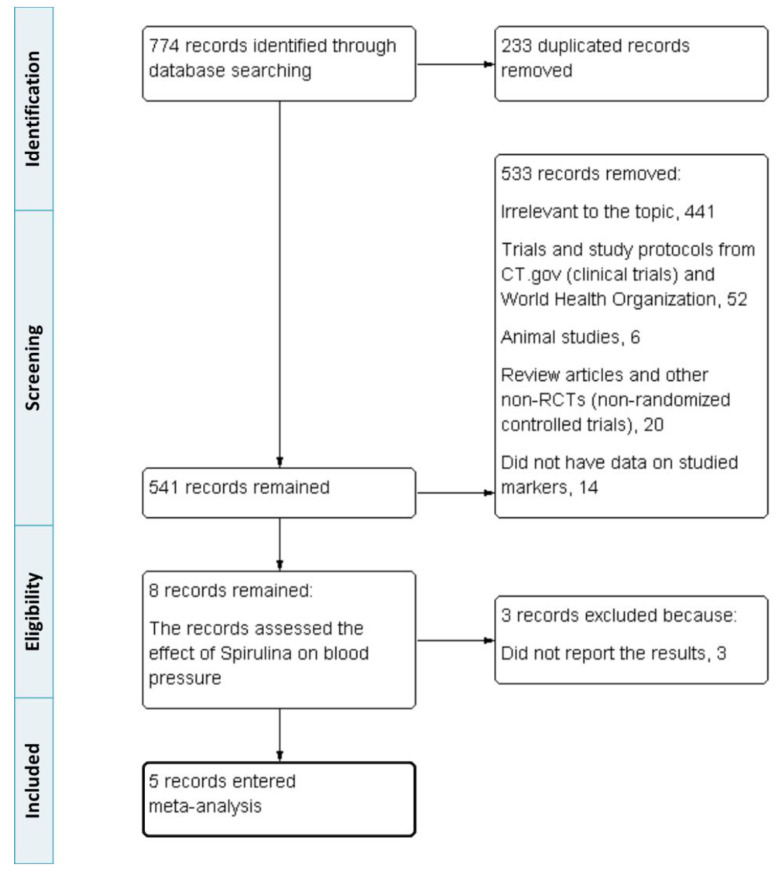
Preferred Reporting Items for Systematic Reviews and Meta-analyses (PRISMA) flow diagram of study identification, inclusion, and exclusion.

**Figure 2 nutrients-13-03054-f002:**
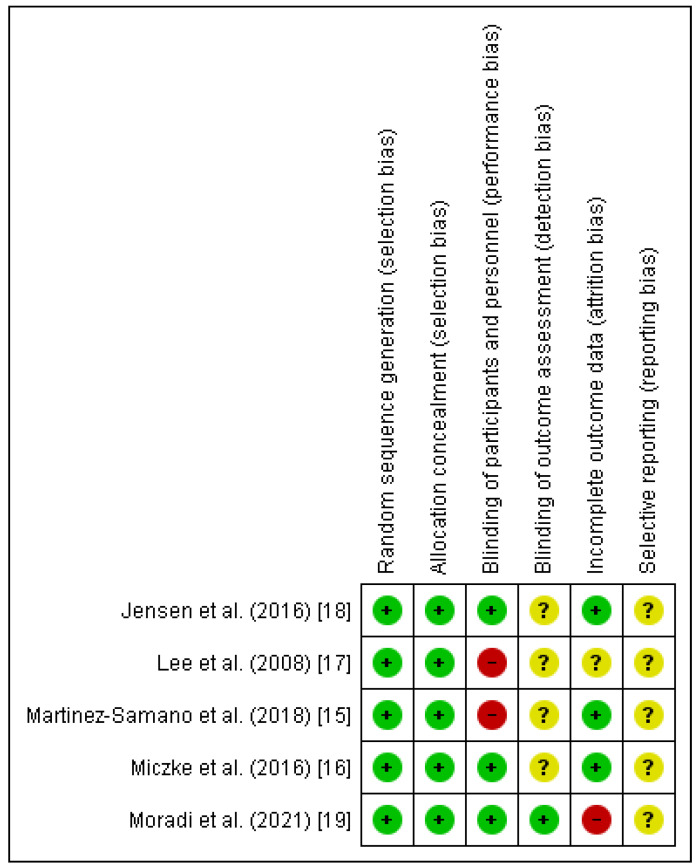
Risk of bias assessment for included randomized controlled trials (RCTs) [15,16,17,18,19].

**Figure 3 nutrients-13-03054-f003:**
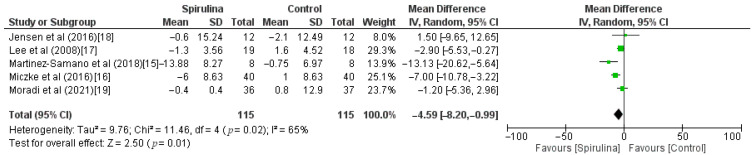
Effect of *Spirulina* supplementation on systolic blood pressure (SBP) compared with the control group [15,16,17,18,19].

**Figure 4 nutrients-13-03054-f004:**
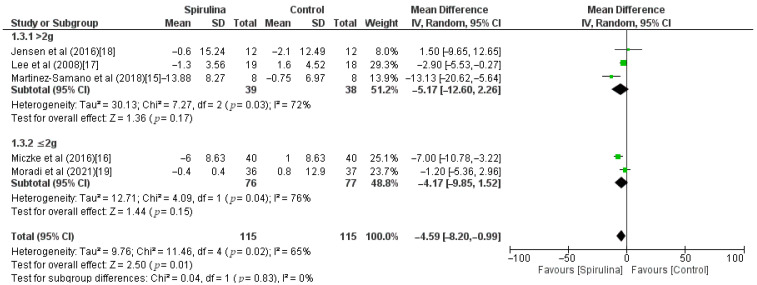
Effect of *Spirulina* supplementation on systolic blood pressure (SBP) compared with the control group stratified by *Spirulina* dosage [15,16,17,18,19].

**Figure 5 nutrients-13-03054-f005:**
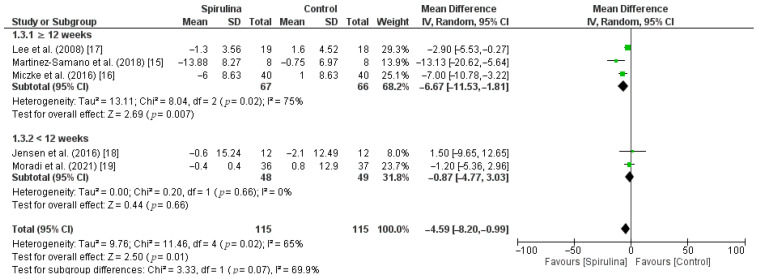
Effect of *Spirulina* supplementation on systolic blood pressure (SBP) compared with the control group stratified by duration of *Spirulina* intake [15,16,17,18,19].

**Figure 6 nutrients-13-03054-f006:**
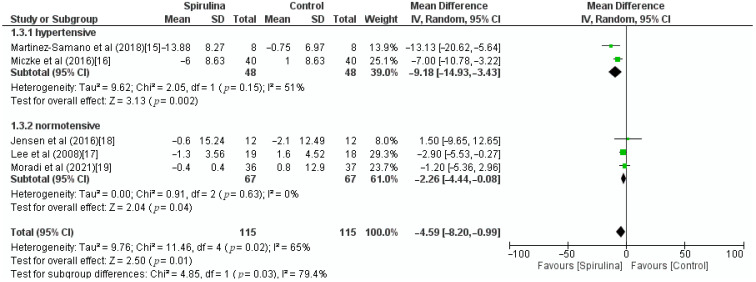
Effect of *Spirulina* supplementation on systolic blood pressure (SBP) compared with the control group stratified by baseline blood pressure [15,16,17,18,19].

**Figure 7 nutrients-13-03054-f007:**
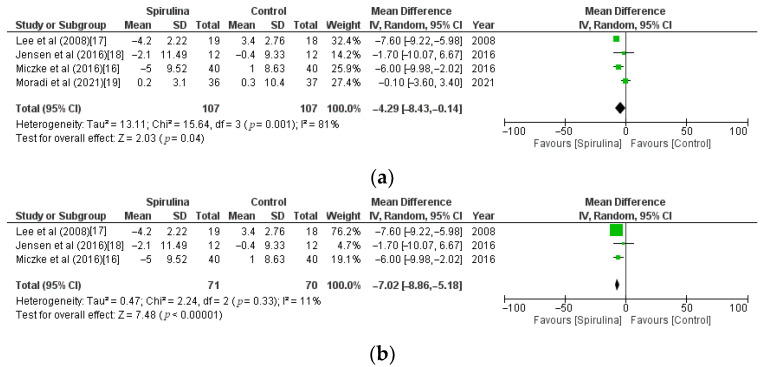
(**a**). Effect of *Spirulina* supplementation on systolic blood pressure (SBP) compared with the control group stratified by baseline blood pressure [16,17,18,19]. (**b**). Effect of *Spirulina* supplementation on diastolic blood pressure (DBP) compared with the control group—studies causing high heterogeneity excluded [16,17,18].

**Table 1 nutrients-13-03054-t001:** Formulae used for data extraction for continuous outcomes.

Name	Formulae
Formula for combining groups—mean	Mean = N_1_M_1_ + N_2_M_2_/N_1_ + N_2_
Formula for combining groups—SD	SD = square root [(N_1_ − 1)SD_1_^2^ + (N_2_ − 1)SD_2_^2^ + (N_1_N_2_/N_1_ + N_2_) × (M_1_^2^ + M_2_^2^ − 2M_1_M_2_)]/ (N_1_ + N_2_ − 1)
Formula for imputing correlation coefficient	Corr = (SD_E_^2^ + SD_C_^2^ − SD_diff_^2^)/(2 × SD_E_ × SD_C_)

N—sample size, M—mean, SD_E_—standard deviation of experimental intervention, SD_C_—standard deviation of control intervention, SD_diff_—difference between experimental and control interventions’ SD.

**Table 2 nutrients-13-03054-t002:** Participants, Intervention, Comparison, Outcomes and Study design (PICOS) criteria for inclusion and exclusion of studies.

Parameter	Description
Participants	Adults with arterial hypertension, diabetes mellitus type 2, Body Mass Index (BMI) < 35 and chronic joint pain, or ulcerative colitis
Interventions	Oral daily supplementation of *Spirulina* (1 g, 2 g, 2.3 g, 4.5 g, 8 g) for 2, 8, or 12 weeks
Comparison	Oral daily placebo (pure microcrystalline cellulose or not specified) or no intervention
Outcomes	Decrease of SBP and DBP
Study design	Parallel RCTs were included in the meta-analysis

**Table 3 nutrients-13-03054-t003:** Detailed parameters of systolic blood pressure (SBP) and diastolic blood pressure (DBP) from the included randomized controlled trials (RCTs).

Reference	No. of Subjects	Outcomes Studied before and after Supplementation	*Spirulina* Group	Placebo/Non-*Spirulina* Group
Martínez-Sámano et al., 2018 [15]	*n* = 8 *Spirulina*, *n* = 8 placebo	SBP (mmHg) before	140.38 ± 9.04	140.75 ± 7.03
		SBP (mmHg) after	126.50 ± 5.53	140.00 ± 6.05
		DBP (mmHg) before	83.75 ± 5.31	84.25 ± 5.28
		DBP (mmHg) after	-	-
Miczke et al., 2016 [16]	*n* = 40 *Spirulina*, *n* = 40 placebo	SBP (mmHg) before	149 ± 7	150 ± 7
		SBP (mmHg) after	143 ± 9	151 ± 9
		DBP (mmHg) before	84 ± 9	85 ± 9
		DBP (mmHg) after	79 ± 9	86 ± 7
Lee et al., 2008 [17]	*n* = 19 *Spirulina*, *n* = 18 non-Spirulina	SBP (mmHg) before	130.7 ± 3.8	131.8 ± 4.0
		SBP (mmHg) after	129.4 ± 2.7	133.4 ± 4.5
		DBP (mmHg) before	84.0 ± 2.1	80.1 ± 2.5
		DBP (mmHg) after	79.8 ± 2.1	83.5 ± 2.7
Jensen et al., 2016 [18]	*n* = 12 *Spirulina* (female F = 10, male M = 2), *n* = 12 placebo (F = 9, M = 3)	SBP (mmHg) before	F 120.8 ± 15.3 M 113.5 ± 6.4	F 112.7 ± 8.9 M 131.7 ± 5.5
		SBP (mmHg) after	F 120 ± 15.8 M 114 ± 2.8	F 111.3 ± 9.9 M 127.7 ± 9.0
		DBP (mmHg) before	F 75.5 ± 12 M 75 ± 5.7	F 70.8 ± 9.1 M 75 ± 7
		DBP (mmHg) after	F 73.3 ± 11.6 M 73.5 ± 6.4	F 70.2 ± 9.6 M 75.3 ± 7.6
Moradi et al., 2021 [19]	*n* = 36 *Spirulina*, *n* = 37 placebo	SBP (mmHg) before	118.7 ± 9.0	117.8 ± 16.6
		SBP (mmHg) after	118.3 ± 6.6	118.6 ± 4.8
		DBP (mmHg) before	80.4 ± 5.6	79.7 ± 11.6
		DBP (mmHg) after	80.6 ± 5.3	80.0 ± 4.7

**Table 4 nutrients-13-03054-t004:** Characteristics of included randomized controlled trials (RCTs).

Reference	Location	Quality Score	Study Design	No. of Subjects	Health Status	Intervention Duration	*Spirulina* Group	Control Group	Outcomes Studied
Martinez-Samano et al., 2018 [15]	Mexico	2	Parallel	16	Systemic arterial hypertension	12 weeks	4.5 g per day of *Spirulina maxima*	Placebo	SBP, DBP
Miczke et al., 2016 [16]	Poland	4	Parallel	80	Systemic arterial hypertension	12 weeks	2 g per day of Hawaiian *Spirulina maxima*	Placebo consisted of pure microcrystalline cellulose	SBP, DBP
Lee et al., 2008 [17]	South Korea	3	Parallel	37	Diabetes mellitus type 2	12 weeks	8 g per day of of freeze-dried *Spirulina*	Received nothing (control group was instructed not to take any functional foods or supplements)	SBP, DBP
Jensen et al., 2016 [18]	USA	3	Parallel	24	Adults 25–65 years of age with BMI < 35 and chronic pain related to specific joint(s) for >6 months	2 weeks	2.3 g per day of of phycocyanin-enriched aqueous extract from *Arthrospira* (*Spirulina*) *platensis*	Placebo	SBP, DBP
Moradi et al., 2021 [19]	Iran	5	Parallel	73	Ulcerative colitis	8 weeks	1 g per day of *Spirulina*	Placebo	SBP, DBP

## Data Availability

Not applicable.

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
