# Peer review of "Effect of Spirulina Supplementation on Systolic and Diastolic Blood Pressure: Systematic Review and Meta-Analysis of Randomized Controlled Trials"

_nutrients, 2021, doi:10.3390/nu13093054_

Round 1

Reviewer 1 Report

I read with interest the manuscript entitled ‘ Effect of Spirulina supplementation on systolic and diastolic 2 blood pressure: systematic review and meta-analysis of ran-3 domized controlled trials’. It’s a quite interesting, novel and well present work. A few points should be considered:

  • The inclusion criteria should be mentioned in the manuscript.
  • In the Introduction Section of the manuscript the authors mention ‘Alternatively, natural medical products are mostly used as adjunctive ther-37 apy to improve long-term outcomes in patients with arterial hypertension and to reduce 38 the number of antihypertensive drugs that are taken’. Please add the corresponding reference.
  • The Introduction Section of the manuscript should be shortened. There is no need to mention all the beneficial effects of spirulina. The authors should emphasize on the effect on blood pressure.
  • The authors should comment on the bias risk measurement of the included RCTs.
  • In case of significant heterogeneity which analysis was performed? Fixed or random-effect ?

Author Response

Response to Reviewer 1 Comments

Point 1:  The inclusion criteria should be mentioned in the manuscript.

Response 1: We explained the inclusion criteria and added numbers in the text for more clarity (section 2.2, 3.1). Moreover, the inclusion and exclusion criteria were adopted before the start of the search and were in line with the PICOS model (Table 2).

Point 2: In the Introduction Section of the manuscript the authors mention ‘Alternatively, natural medical products are mostly used as adjunctive therapy to improve long-term outcomes in patients with arterial hypertension and to reduce the number of antihypertensive drugs that are taken’. Please add the corresponding reference.

Response 2: We added the corresponding reference ([5]).

Point 3: The Introduction Section of the manuscript should be shortened. There is no need to mention all the beneficial effects of Spirulina. The authors should emphasize on the effect on blood pressure.

Response 3: We removed unnecessary sentences which were not directly related to the topic of the work.

Point 4: The authors should comment on the bias risk measurement of the included RCTs.

Response 4: We added information about the bias risk measurement according to Cochrane Collaboration’s tool (Figure 2).

Point 5: In case of significant heterogeneity which analysis was performed? Fixed or random-effect?

Response 5: We added information that random effect analysis was performed (section 2.4).

Reviewer 2 Report

The authors performed a meta-analysis and review of publications regarding the effects of Spirulina present in the potential diet on human blood pressure.

The meta-analysis was methodically correct and the minimum number of publications qualified for statistical analysis was reached. The authors pointed out the strong and weak points of the analysis. The results of the meta-analysis were clearly and lucidly presented in the form of tables and graphs. Separate discussion of the results allowed for better understanding of the information obtained from the meta-analysis. The authors provided valuable hints for the organizers of future RCTs (selection of participants in terms of lifestyle and stimulants used, important role of spirulina dose on the obtained effects). The authors explained the principle of spirulina effect on human blood pressure. They pointed out the need of greater control of spirulina-based supplements in terms of health safety.

Conclusions although short are a concise presentation of the most important information arising from the meta-analysis.

I provide some comments below:

  1. In the abstract, I recommend developing abbreviations such as RCT, SBP, DBP, MS, CI, I2, as not all readers may be familiar with what these abbreviations mean.
  2. In study selection section authors wrote that they rejected articles without sufficient data. Please specify what data you are referring to.
  3. The title of section 3.5 is identical to section 3.1. I would also suggest moving the content of section 3.5 to section 3.1 or 3.2.
  4. Section 3.6 is a repetition of information from verses 136-138 and therefore I propose to delete it.
  5. The table from Supplementary materials should be placed in the main body of the article or as an Appendix Table (please check what it is in the manual for authors). This table contains important information for the publication.

Author Response

Response to Reviewer 2 Comments

Point 1: In the abstract, I recommend developing abbreviations such as RCT, SBP, DBP, MS, CI, I2, as not all readers may be familiar with what these abbreviations mean.

Response 1: We developed all the above-mentioned abbreviations.

Point 2: In study selection section authors wrote that they rejected articles without sufficient data. Please specify what data you are referring to.

Response 2: We specified that we rejected articles without data on systolic and diastolic pressure before and after Spirulina intervention (2.2 section).

Point 3: The title of section 3.5 is identical to section 3.1. I would also suggest moving the content of section 3.5 to section 3.1 or 3.2.

Response 3: We moved the content of section 3.5 to section 3.2 and deleted section 3.5.

Point 4: Section 3.6 is a repetition of information from verses 136-138 and therefore I propose to delete it.

Response 4: We removed section 3.6.

Point 5: The table from Supplementary materials should be placed in the main body of the article or as an Appendix Table (please check what it is in the manual for authors). This table contains important information for the publication.

Response 5: We placed the table from Supplementary material to the main body of the article (Table 3).